# Vitamin D Receptor Gene Polymorphism Predicts the Outcome of Multidisciplinary Rehabilitation in Multiple Sclerosis Patients

**DOI:** 10.3390/ijms241713379

**Published:** 2023-08-29

**Authors:** Franca Rosa Guerini, Cristina Agliardi, Letizia Oreni, Elisabetta Groppo, Elisabetta Bolognesi, Milena Zanzottera, Domenico Caputo, Marco Rovaris, Mario Clerici

**Affiliations:** 1IRCCS Fondazione Don Carlo Gnocchi, Via Capecelatro 66, 20148 Milan, Italy; fguerini@dongnocchi.it (F.R.G.); moreni@dongnocchi.it (L.O.); elisabetta.groppo@asst-santipaolocarlo.it (E.G.); ebolognesi@dongnocchi.it (E.B.); mzanzottera@dongnocchi.it (M.Z.); dcaputo@dongnocchi.it (D.C.); mrovaris@dongnocchi.it (M.R.); mario.clerici@unimi.it (M.C.); 2Ospedale San Paolo, ASST Santi Paolo e Carlo, Clinical Neurology Unit, Department of Health Sciences, University of Milan, 20142 Milan, Italy; 3Pathophysiology and Transplantation Department, University of Milan, 20122 Milan, Italy

**Keywords:** multiple sclerosis, vitamin D receptor SNPs, multidisciplinary rehabilitation

## Abstract

Better knowledge about the possible role of genetic factors in modulating the response to multiple sclerosis (MS) treatment, including rehabilitation, known to promote neural plasticity, could improve the standard of care for this disease. *Vitamin D receptor* (*VDR*) gene polymorphisms are associated with MS risk, probably because of the role played by vitamin D in regulating inflammatory and reparative processes. The aim of this study was to evaluate the association of the most important functional *VDR* SNPs (*TaqI* (*T*/*C*), *ApaI* (*A*/*C*), and *FokI* (*C*/*T*)) with functional outcome in MS patients undergoing multidisciplinary inpatient rehabilitation (MDR) treatment, in order to determine whether genetic profiling might be useful to identify subjects with a higher chance of recovery. To this end, 249 MS inpatients with a diagnosis of either progressive (pMS; n = 155) or relapsing remitting (RRMS; n = 94) disease who underwent MDR treatment (average duration = 5.1 weeks) were genotyped for *VDR* SNPs by real-time allelic discrimination. The rehabilitation outcome was assessed using the modified Barthel Index (mBI), Expanded Disability Status Scale (EDSS), and pain numerical rating scores (NRS) at the beginning and the end of MDR treatment. A positive correlation was observed in RRMS patients between the *VDR TaqI* major allele (TT) and mBI increase (i.e., better functional recovery), as assessed by the linear and logistic regression analysis adjusted for gender, age, disease duration, time of hospitalization, *HLA-DRB1*15.01* positivity, and number of rehabilitative interventions (Beta = 6.35; *p* = 0.0002). The *VDR-1 TaqI*, *ApaI*, *FokI*: TCC haplotype was also associated with mBI increase in RRMS patients (Beta = 3.24; *p* = 0.007), whereas the *VDR-2*: CAC haplotype was correlated with a lower mBI increase (Beta = −2.18 *p* = 0.04) compared with the other haplotypes. *VDR TaqI* major allele (TT), as well as the *VDR-1 TaqI*, *ApaI*, *FokI*: TCC haplotype could be associated with a better rehabilitation outcome in RRMS patients.

## 1. Introduction

Multiple sclerosis (MS) is a demyelinating disease of the central nervous system. MS can be clinically categorized as primary progressive MS (PPMS) or relapsing-remitting MS (RRMS); RRMS may evolve over time into secondary progressive MS (SPMS) [1]. The identification of biomarkers that could help to clarify MS pathogenesis and pathophysiology, predict disease progression, and be used as a paraclinical tool for treatment choice and monitoring, has proven to be a difficult task. This is mostly as a consequence of the interplay between inflammation, demyelination, and neurodegeneration, which is the basis of the disease [2] and results in physical, cognitive and psychological disabilities.

The ultimate cause of MS is still unknown, but a combination of demographic, environmental, and genetic factors has been associated both with the risk of developing MS and with different MS clinical phenotypes [3]. Genetic factors, including human leukocyte antigen (*HLA*) genes, and interleukin 2 and 7 receptor (*IL2R* and *IL7R*) gene polymorphisms, have been identified as MS risk factors [4]. *HLA-DRB1*15.01*, in particular, shows the strongest association with MS, principally due to its role in autoimmunity [5]. Notably, the interaction between host genetic background and environmental risk factors, including Epstein–Barr (EB) virus infection and vitamin D (VD), has been suggested to play a crucial role in MS risk and development [6,7,8].

The likelihood of developing MS has been suggested to be determined at conception, when protective and deleterious genetic factors, including *HLA-DRB1*15.01* and genes involved in VD metabolism, are inherited [9]. Notably, a VD responsive element (VDRE) sequence is present in the promoter region of the *HLA DRB1*15* gene; this suggests a role of the VD/vitamin D receptor (VDR) axis in regulating the transcription of this HLA molecule, possibly bridging MS environmental and genetic risk factors [10,11,12]. An analysis of all of the possible phenotypic combinations of the *VDR* genotype and the *HLA DRB1*15.01* allele in a cohort of MS patients and healthy controls showed that the co-presence of *VDR TaqI* (TT) in subjects carrying the risk factor *HLA DRB1*15.01* (i.e., *DRB1*15.01* positive) was indeed significantly more frequent in healthy subjects compared with MS patients, suggesting that the *VDR TaqI* (TT) genotype may exert a protective role for MS [11].

From a pathogenic perspective, possible explanations for the association between MS risk and VD include the role of vitamin D in immunomodulatory [13] and remyelination processes [14,15,16,17].

In this context, the *VDR* gene, which mediates both genomic and non-genomic VD effects, is strongly suspected to play a pivotal pathogenic role. The *VDR* gene, located on chromosome 12.13.11, encompasses different single nucleotide polymorphisms (SNPs), the most important of which are *ApaI*, *BsmI*, *TaqI*, and *FokI*, which have been shown to be associated with autoimmune disorders including MS [8]. *ApaI* (rs7975232 *A* > *C*) [18] and *BsmI* (rs1544410 *C* > *T*) [19] are located in intron 8, whereas *TaqI* (rs731236 *T* > *C*) is responsible for a silent mutation in exon 9 [20], and *FokI* (rs2228570 *C* > *T*) is a missense mutation located in exon 2. *FokI*, in particular, modifies the length of the protein, resulting in a VDR protein with either 427 (*T* variant) or 424 amino acids (*C* variant), with the latter being biologically more active [21].

Multidisciplinary rehabilitation (MDR) could be defined as a coordinated intervention, delivered by two or more disciplines based on an interdisciplinary evaluation assessment of patient needs, which aims to help persons in achieving and maintain maximal physical, psychological, social, and vocational potential and quality of life (QoL) consistent with impairment, environment, and life goals. In principle, rehabilitation interventions for persons with MS may include exercise, functional training, equipment prescription, provision of assistive technology, orthotics prescription, teaching of compensatory strategies, caregiver/family support and education, counselling and referral to community resources. These can be targeted for a variety of impairments, such as mobility, fatigue, pain, dysphagia, bladder/bowel dysfunction, decreased independence in activities of daily living (ADL), communication, QoL, affective disorders, and cognitive dysfunction. We previously reported that inpatient MDR including multidomain interventions (motor, cognitive, speech, etc.) tailored to individual subject needs and goals improves autonomy and daily living activities in a relevant proportion of MS patients, with greater efficacy in those patients with an RR disease phenotype and shorter disease duration [22]. Research protocols are underway to better understand the possible correlations between genetic background and response to therapy in MS patients; however, few protocols investigate the role of genetic risk factors in modulating the response to rehabilitation treatment. To the best of our knowledge, the relationship of functional recovery with genetic profile in MS has been investigated only in two studies focused on *BDNF* Val66Met polymorphism [23] and on cannabinoid receptor polymorphisms [24].

In this retrospective study, we analyzed data from MS patients who underwent MDR, consisting of daily sessions (Monday-to-Saturday) for at least 500 min/week. The rehabilitation program was based on multidisciplinary activities that could include physiotherapy, occupational therapy, respiration therapy, cognitive rehabilitation, speech and swallowing rehabilitation, physical therapy for pain and formal psychological counselling, assessing patient needs and goals, performed by a neurologist together with a physical medicine and rehabilitation specialists [22]. Patients were genetically characterized for *HLA-DRB1*15.01* and *VDR TaqI*, *BsmI*, *ApaI*, and *FokI* polymorphisms to verify whether different *VDR* polymorphisms could represent a prognostic marker of rehabilitative outcome.

The ultimate aim was to ascertain whether such genetic profiling could be useful in identifying subjects with a higher chance of recovery at the end of the MDR program.

## 2. Results

### 2.1. VDR Polymorphism Distribution and Disability Indexes in MS Patients

The *VDR TaqI*, *BsmI*, *ApaI*, and *FokI* genotype distribution in MS patients is reported in Table 1. The distribution of all *VDR* polymorphisms, with the exception of the *BsmI* genotype (*p* < 0.01), conformed to the Hardy–Weinberg equilibrium. Thus, as we could not exclude that this disequilibrium could be due to a problematic genotype assay [25], the *BsmI* polymorphism was excluded from all of the subsequent analyses to avoid introducing possible correlation bias.

As PPMS and SPMS patients were clinically, demographically, and genetically homogenous, for subsequent analyses, MS patients were grouped as progressive MS (PMS), including both these phenotypes, and RRMS.

Overall, the Expanded Disability Status Scale (EDSS), modified Barthel Index (mBI), and pain numerical rating score (NRS) values collected upon admittance (T0) and after MDR (T1) were similar to those observed in the larger group studied by Groppo et al. [22]. The EDSS, mBI, and pain NRS scores were not normally distributed after the Kolmogorov Smirnov evaluation test; thus, their values were reported as median and interquartile range (IQR), and the non-parametric Mann–Whitney comparison was adopted to evaluate the differences between PMS and RRMS patients at both T0 and T1 points. Statistically higher EDSS scores were observed in PMS compared with RRMS patients at both T0 (median: 7 IQR: 1.5 vs. median 6 IQR: 1.5, respectively) (*p* < 0.001) and T1 (median: 6.5 IQR: 1.3 vs. median: 6 IQR: 1.5 respectively) (*p* < 0.001) (Figure 1A). On the other hand, PMS patients had lower mBI scores than RRMS at both T0 (median: 57 IQR: 30.5 vs. median: 75 IQR: 14.0) (*p* < 0.0001) and T1 (median: 65.0 IQR: 29.5 vs. median: 83.0 IQR: 11.8) (*p* < 0.0001) (Figure 1B), confirming a higher disability progression in PMS patients than in RRMS patients. Conversely, no difference was observed for pain NRS scores between groups (Figure 1C).

The analysis of repeated measures for EDSS, mBI, and pain NRS scores at T0 and T1 are shown in Table 2. The results showed a significant improvement for all of the outcome indicators after MDR in the whole group of patients (Total row) *p* < 0.001, and in patients grouped according to disease phenotype (PMS and RRMS rows) *p* < 0.001. No difference was observed between *DRB1*15.01* positive and negative patients.

### 2.2. Disability Indexes and MDR Outcome Correlate with VDR TaqI–ApaI–FokI Polymorphisms

The genetic *VDR* pattern was evaluated in relation to the EDSS, mBI, and NRS values at admittance and with their changes after rehabilitation (Appendix A), repeated measures analysis showed a significant variation of mBI in relation to *TaqI* (*p* < 0.001) and *ApaI* genotypes (*p* = 0.01). Specifically, a higher delta mBI median value (i.e., better functional recovery) was observed in RRMS patients carrying the major allele *TaqI* (TT) (median: 11.0, IQR: 9.0) than in those patients carrying the heterozigous genotype (TC) (median: 4.5, IQR: 5.8 *p* < 0.001) or the homozygous minor allele (CC) (median: 6.0, IQR 5.5; *p* = 0.001) (Figure 2(A1)).

Similarly, a higher delta mBI value was observed in patients carrying the homozigous minor allele *ApaI* (CC) (median: 11.5, IQR: 5.8) genotype than in those carrying the heterozigous (AC) (median: 6.0, IQR: 7.0 *p* = 0.02) and the homozigous major allele (AA) (median: 5.5, IQR: 9.0 *p* = 0.005) genotypes (Figure 2(A2)). In both cases, the *TaqI* dominant (DM) (TT vs. TC + CC) and the *ApaI* recessive models (RM) (CC vs. CA + AA) were associated with mBI outcome. No correlations were observed between *FokI* genotypes and disability indexes (Figure 2(A3)).

Notably, and once again underlining the profoundly different immunological milieu that is observed in RRMS and PMS, no significant association of *VDR TaqI*, *ApaI*, and *FokI* genotypes with clinical rehabilitation outcomes was observed in PMS patients (Figure 2(B1–B3)).

To take into account all of the factors that could influence MDR outcome, a general linear analysis of delta mBI, EDSS, and pain NRS values with *VDR TaqI* DM (TT vs. TC + CC), *ApaI* (RM) (CC vs. AC + AA), and *FokI* genotypes was performed, adjusting all of the analyses for each basal value and for the following indexes: gender, *DRB1*15.01* positivity, age, disease duration (years), days of hospitalization, and number of interventions. A significant association of higher delta mBI was found with *TaqI* (TT) (Beta = 6.35 *p* < 0.0002), with lower age (Beta = −0.21 *p* = 0.003), and lower mBI at T0 (Beta = −0.23 *p* = 0.0002) in RRMS patients alone. No associations with *ApaI* and *FokI* polymorphisms were observed (Table 3). No correlations were detected between delta EDSS, pain NRS, and *VDR* polymorphisms.

To further assess the association between delta mBI and *VDR* polymorphisms, delta mBI was categorized as a binary variable based on a minimum discharge improvement of 5 points [22] (Delta mBI ≥ 5 vs. Delta mBI < 5). This variable was then used as the dependent variable in the binary logistic regression analysis, and its correlation with *TaqI* DM, *ApaI*, RM, and *FoqI* genotypes was considered. All of the correlations were adjusted for basal values of mBI (T0), age, gender, *DRB1*15.01* positivity, number of interventions, disease duration, and days of hospitalization. A significant association was found between delta mBI ≥ 5 and the homozygous *TaqI* (TT) genotype (*p* = 0.01; OR: 7.92, 95% CI: 1.5–40.9), lower age (*p* = 0.01 OR: 0.94, 95% CI: 0.89–0.98), and lower mBI at T0 (*p* = 0.03, OR: 0.94, 95% CI: 0.89–0.99), once again only in the RRMS patients.

Both the linear and the logistic regression analysis confirmed that the mBI outcome after MDR was dependent on age, basal mBI, and *VDR TaqI* genetic background.

### 2.3. Disability Indexes and MDR Outcome Correlation with VDR TaqI–ApaI–FokI Haplotype Analysis

The haplotype analysis was performed next; the results confirmed the presence of a *VDR TaqI–ApaI* polymorphism linkage disequilibrium in patients (R^2^ = 0.49; Figure 3).

The haplotype distribution comparison analysis did not show statistical differences between PMS and RRMS patients. A quantitative haplotype trait analysis was carried out to evaluate the *VDR TaqI–ApaI–Fok1* haplotype correlation with delta mBI, delta EDSS, and delta NRS in RRMS and PMS patients. The *VDR-1*: TCC haplotype was associated with a higher delta mBI (Beta 3.24; *p* = 0.007), whereas the *VDR-2*: CAC haplotype was associated with a lower Delta mBI (Beta = −2.18; *p* = 0.04) in RRMS patients only (Table 4).

Finally, no significant associations were observed between haplotype distribution and either delta EDSS or pain NRS values.

## 3. Discussion

In this study, we investigated whether *VDR* gene polymorphisms could be considered as candidate biomarkers associated with the functional outcome after MDR rehabilitation in RRMS and PMS patients. To this end, four of the best characterized *VDR* polymorphisms, *TaqI*, *BsmI*, *ApaI*, and *FokI*, were evaluated in a large group of MS subjects who underwent MDR and who were fully characterized for EDSS, mBI, and pain NRS clinical scores.

Firstly, clinical characteristics and *VDR* polymorphisms distribution were compared in MS patients. As the *BsmI* genotype distribution did not respect the Hardy–Weinberg equilibrium in the entire population studied, this genotype was discarded from all further association analyses [25]. Furthermore, because *HLA-DRB1*15.01* is the main genetic risk factor in MS development, and its expression is regulated by VDR, all of the subjects were characterized for *HLA-DRB1*15.01* and clustered on the basis of positivity, finding that *HLA-DRB1*15.01* did not correlate with the outcome of MDR.

Conversely, a direct correlation between the *VDR TaqI* (TT) major allele and the *ApaI* (CC) minor allele homozigous genotypes was observed with MDR-associated functional independence improvement, as measured by mBI in RRMS patients alone. To better understand whether the *VDR* genetic background plays a role in rehabilitation outcomes, linear and logistic regression analysis were used next to evaluate *VDR TaqI*, *ApaI*, and *FokI* association with delta mBI, delta EDSS, and delta pain NRS outcomes after MDR, taking into consideration age, gender, disease duration, days of hospitalization, number of interventions, and *DRB1*15.01* positivity, as well as each basal value, as covariates, which may also have an impact on MDR outcome. Both these approaches confirmed the presence of a strong *TaqI* (TT) association with increased mBI, thus with a clearly improved rehabilitation outcome after MDR; this effect was more evident in younger RRMS patients with a lower mBI upon admittance. Notably, *ApaI* (CC) association with MDR outcome lost statistical significance after multi comparative correction.

No correlations between *VDR* genotype distribution and rehabilitation outcome were observed in PMS patients, and no associations were observed between any genetic parameter and EDSS or pain NRS changes after MDR, either in RRMS or in PMS patients. Finally, the haplotype analyses of *VDR* SNPs confirmed the presence of a linkage disequilibrium between *TaqI–ApaI* polymorphisms in MS patients.

The haplotype distribution was evaluated both in RRMS and in PMS patients, and no differences were found between the two groups of patients. However, haplotype correlation with MDR outcomes indicated the presence of a significant association, with a higher mBI delta score in RRMS carrying the *TaqI-ApaI-Fok1 VDR-1*: TCC haplotype. Conversely, the *VDR-2*: CAC haplotype was associated with a lower increase in mBI, i.e., a less favorable effect of the MDR treatment. It is evident that these two haplotypes were genetically complementary for *TaqI* and *ApaI* polymorphisms, whereas no impact of *FokI* was evidenced.

Consistent with our previous report on a larger sample of subjects [22], including the subgroup genetically analyzed for the present study, inpatient MDR was associated with improved autonomy in activities of daily living in a relevant proportion of patients with MS. Moreover, an association between *VDR* genetic background and positive MDR outcome was found, although it is worth noting that any interpretation of our results for the purpose of individual patient profiling for rehabilitation programs is premature. Indeed, due to the characteristics of our inpatient MDR treatment, the activity undergone by each patient was heterogeneous, as it was based on a personalized and not on a pre-defined study protocol. Conversely, the latter was the case in other studies [23,24] investigating the role of genetic biomarkers in MS rehabilitation. Giordano et al. [23] studied MS subjects undergoing inpatient motor rehabilitation and found an association between *BDNF* genetic variants and the improvement in two clinical scales assessing ambulation and hand dexterity. On the other hand, Mori et al. [24] found an association between *CB1* receptor polymorphisms and poor clinical benefit after physical therapy in an outpatient setting. Therefore, to the best of our knowledge, our findings may provide a novel piece of evidence on the potential biomarkers predicting individualized MDR efficacy on functional autonomy in patients with MS.

The association of *VDR* SNPs with MS risk has been extensively analyzed and conflicting results have been reported. Some studies did not observe any involvement of *VDR ApaI*, *TaqI*, *BsmI*, and *FokI* in Caucasian MS patients [26], but suggested *ApaI* (A) allele as a risk factor in Asian populations. Other analyses showed the same *ApaI* (A) allele to be a protective factor [27] against MS development. Importantly, a protective role of the *TaqI* (TT) genotype was repeatedly shown to be present both in Asian and Caucasian MS patients [28,29], and a study in Italian MS patients showed a protective role against MS risk for *VDR TaqI* (TT) subjects expressing *HLA-DRB1*15.01* [11].

Our results are also consistent with the hypothesis that *TaqI* plays a role in the pathogenesis of MS [8]. More importantly, the data herein indicate a role of *TaqI* (TT) in predicting a better outcome of rehabilitation in RRMS patients, as assessed by the mBI scale. The results of the haplotype analysis evidencing the involvement of both *TaqI* (T) and *ApaI* (C) within a protective *VDR-1*: TCC haplotype may also explain the discordant results on *ApaI* association with MS, as *ApaI* involvement may be the result of linkage disequilibrium with *TaqI*. Finally, our data agree with evidence indicating that the *VDR FokI* polymorphism is not involved in MS development [26,27], although not every result agrees with this conclusion [30,31].

No correlations were observed between *VDR* polymorphisms and rehabilitation outcome in PMS patients. This finding again underlies the profoundly different immunological milieu that is observed in RRMS and PMS. In this context, it is important to underline that vitamin D supplementation and immunomodulatory treatments have been shown to not be efficient when the progression of disability is already clinically evident, i.e., in the progressive forms of disease [9].

A limit of this retrospective study is that we could not measure vitamin D concentration and its relation with MDR outcome. Importantly, though, (1) *TaqI* polymorphism was shown to play a role in improving the stability of *VDR* mRNA and protein translation efficiency [32], and (2) the *TaqI* (TT) genotype was observed to be associated with higher serum vitamin D levels in MS patients [30]. Taken together, these results suggest that vitamin D concentration is increased in *TaqI* (TT) RRMS patients, which are characterized by a better response to MDR.

The complexity of MS requires a comprehensive approach for the patient, which is based on both pharmacological and nonpharmacological interventions [33], and rehabilitation remains a mainstay across various stages of this condition. Typically, rehabilitation requires a personalized approach for the patient and for modifications of the environment, as it works according to levels of disability, with interventions varying on the basis of the remaining abilities [34,35,36,37]. It also aims at improving functional independence [38]. For these reasons, adding paraclinical information, such as genetic profiling, to better define intervention programs and priorities might represent a rewarding strategy.

## 4. Materials and Methods

### 4.1. Patients

Patients affected by MS (diagnosed according to McDonald’s revised diagnostic criteria [39] who were admitted to the Neurorehabilitation Unit, MS Center, Scientific Institute Don Gnocchi (Milan), for hospital-based treatment (see [22]) were genetically characterized to determine possible genetic predictors of treatment outcome. Among 655 patients participating to the original study [22], 249 were selected for the present study; informed consent was obtained from all individuals prior to inclusion in the study. The study was conducted according to the guidelines of the Declaration of Helsinki and was approved by the institutional review board of the Don Carlo Gnocchi ONLUS Foundation, Milan (Protocol number #11_27/06/2019). Patients’ clinical and demographic data are summarized in Table 5.

### 4.2. Rehabilitation Treatment

Admission criteria for MDR were the presence of two or more moderate neurological disabilities upon clinical evaluation and recent (i.e., within 6 months) functional deterioration. The intensive rehabilitation program included physiotherapy (i.e., motor rehabilitation) in all patients, associated with occupational therapy (in 78.3% of them), speech and swallowing rehabilitation (55.0%), cognitive rehabilitation (22.9%)., respiration therapy (16.5%), formal psychological counselling (28.5%), and physical therapy for pain (e.g., massage therapy, transcutaneous electrical nerve stimulation, electrical stimulation, ionophoresis) (89.1%). When needed, additional evaluations (cognitive, urological, ophthalmological, respiratory, etc.) were performed by the single discipline specialists to define the program. MDR consisted of daily individual sessions of one or more activities from Monday to Saturday for a total of at least 500 min a week; the duration of the admission was established following an intermediate multidisciplinary re-assessment of the program and goals with the involvement of physicians, therapists, and nurses after two to three weeks of admission. The Modified Barthel index (mBI) score, Expanded Disability Status Scale (EDSS), and pain numerical rating score (NRS) were rated upon admission (T0) and at discharge (T1) [22].

### 4.3. Samples Collection and DNA Extraction

Whole blood was collected in EDTA-containing vacutainer tubes (Becton Dickinson Co., Rutherford, NJ, USA); genomic DNA was extracted from peripheral blood mononuclear cells (PBMC) using a standard phenol/chloroform procedure. The DNA amount for each sample was determined by measuring the optical density at 260 nm wavelengths using a spectrophotometer (SmartSpec Plus, Bio-Rad, Irvine, CA, USA). The DNA samples were stored at −20 °C until use.

### 4.4. HLA-DRB1*15.01 Characterization

The presence of the *HLA-DRB1*15.01* allele, either in heterozygous or homozygous form, was inferred by the genotyping of the tag SNP rs3135388 [40] by allelic discrimination real-time PCR with the TaqMan™ C__27464665_30 probe and following the procedure described above.

### 4.5. VDR Polymorphisms and Genotyping

The following *VDR* polymorphisms were analyzed rs731236 T/C(aka *TaqI*), rs1544410 C/T (*BsmI*), rs7975232 A/C (*ApaI*), and rs2228570 C/T (*FokI*). These SNPs were selected based on their positions in the *VDR* gene or for their potential functional role, and were evaluated by allelic discrimination real-time PCR using pre-designed TaqMan probes (Applied Biosystems, Foster City, CA, USA). PCR consisted of a hot start at 95 °C for 10 min followed by 40 cycles of 94 °C for 15 s and 60 °C for 1 min. Fluorescence detection took place at 60 °C. Assays were performed in 10 μL reactions, using a TaqMan Genotyping Master Mix on 96-well plates using an ABI 7000 instrument (Applied Biosystems). Control samples representing all possible genotypes and a negative control were included in each reaction.

### 4.6. Statistical Analysis

Chi-square analysis was applied to both verify that populations were in Hardy–Weinberg equilibrium (HWE) and to evaluate *VDR* SNPs differences between disease phenotype groups by evaluating the genotype distribution. Two-sided *p* values after Bonferroni’s adjustment (*p*_c_) for multiple comparisons were calculated and the significance threshold was set at *p* < 0.05.

As the Kolmogorov–Smirnov analysis evidenced that numerical data were not normally distributed, age, disease duration, and duration of MDR treatment, as well as mBI, EDSS, and pain NRS scores were reported as median and interquartile range (IQR). mBI, EDSS, and pain NRS improvements after MDR were reported as delta score and statistical analysis applied, as in [22].

Non-parametric Kruskal–Wallis and Mann–Whitney tests were applied to evaluate mBI, EDSS, and pain NRS scores’ correlation with the phenotype groups, as well as with *HLA-DRB1*15.01* positivity, at the admittance (T0) and after MDR (T1), and to evaluate Delta mBI, EDSS, and pain NRS scores’ correlation with *VDR* genotype distribution in each phenotype group. Wilcoxon signed-rank test for repeated measures was applied to compare T0 and T1 score overall and in phenotype groups.

A general linear regression model was than applied in RRMS and PMS, separately, considering Delta mBI, delta EDSS and delta pain NRS as dependent variables, which have been correlated with *VDR TaqI* DM (TT vs. TC + CC), *ApaI* (RM) (CC vs. AC + AA), and *FokI* genotypes. In the same model mBI, EDSS, and pain NRS scores at the baseline (T0), as well as age, years of disease duration, days of hospitalization, and number of interventions were included as numerical covariates, whereas *VDR* genotypes, *DRB1*15.01* positivity, and gender were imputed as categorical variables.

Binary logistic regression analysis was used to confirm whether the *VDR* Dominant Model *TaqI* (TT vs. TC + CC) polymorphisms may be predictors of MDR effects on improvement of mBI, defined as an increase of at least 5 points at discharge (22): Delta mBI ≥ 5 vs. Delta mBI < 5 was imputed as a dependent variable, while mBI at T0, age, gender, *DRB1*15.01* positivity, *VDR TaqI* Dominant Model, *ApaI* Recessive Model (CC vs. AC + AA), *FokI* genotypes, number of interventions, years of disease duration, and days of hospitalization were taken into account again as covariates. Statistical analyses were performed using SPSS software (v.28, IBM, in Armork, NY, USA) and SAS software (v. 9.4; SAS Institute Inc., Cary, NC, USA).

Finally, a haplotype analysis of distribution in MS patients’ groups was performed, and the haplotype correlation with delta values of mBI, EDSS, and NRS was evaluated by quantitative trait analysis. To this end, SHESIS plus online software (http://analysis.bio-x.cn (accessed on 1 June 2023)), was adopted [41,42], that use an entropy-based algorithm to detect epistasis in the context of quantitative trait datasets [43].

## 5. Conclusions

Although the results need to be confirmed in larger cohorts of patients, they are the first to suggest a direct correlation between *VDR* polymorphisms and rehabilitative outcome. These results also suggest a possible use of patients’ *VDR* genetic background in custom-tailoring the rehabilitative approach to disease, in the light of a precision medicine-based approach.

## Figures and Tables

**Figure 1 ijms-24-13379-f001:**
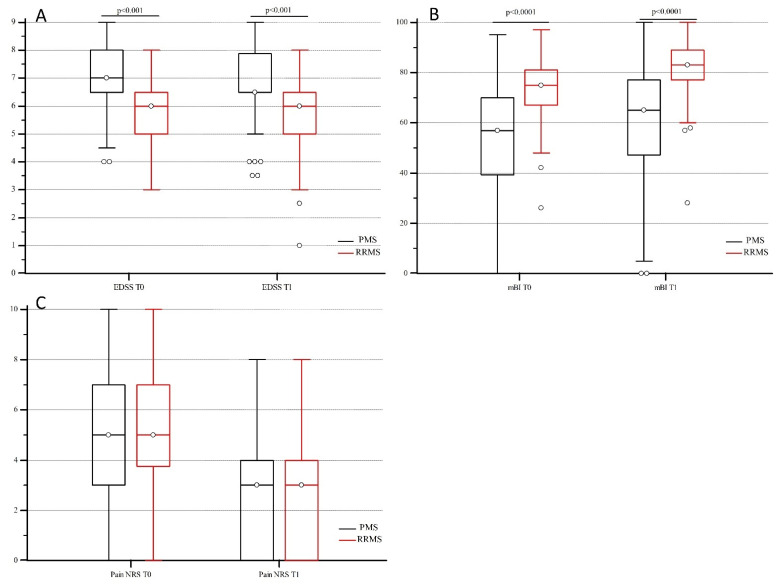
EDSS (**A**), mBI (**B**), and pain NRS (**C**) score distribution in 155 PMS (black line) and 94 RRMS (red line) patients upon admittance (T0) and after MDR treatment (T1). Median scores are reported and a Mann–Whitney comparison is applied for not normal distribution. Significant differences of EDSS (*p* < 0.001) (**A**) and mBI scores (*p* < 0.0001) (**B**) at T0 and T1 are evidenced between PMS and RRMS. EDSS: Expanded Disability Status Scale; mBI: Modified Barthel index; NRS: pain numerical rating score.

**Figure 2 ijms-24-13379-f002:**
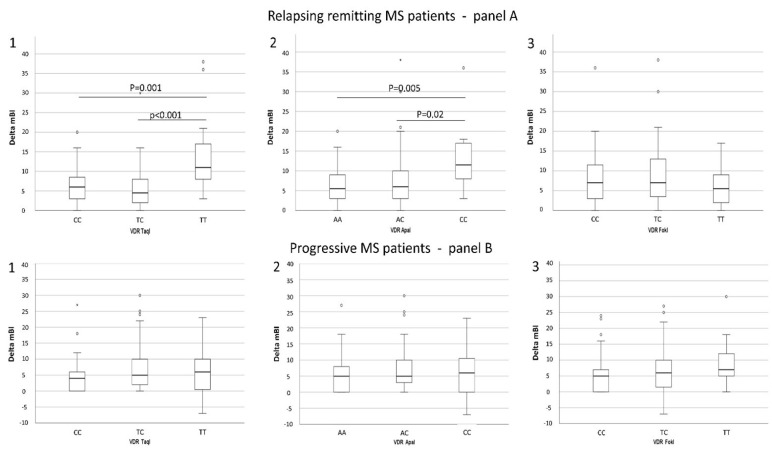
Delta mBI median distribution in relation to *VDR TaqI*, *ApaI*, and *FokI* genotypes in 94 RRMS (**A1**, **A2**, **A3** respectively) and 155 PMS (**B1**, **B2**, **B3** respectively) patients. Higher delta mBI was observed in RRMS patients carrying the *TaqI* (TT) genotype than in those with the *TaqI* (CT) (*p* < 0.001) and *Taq* (CC) (*p* = 0.001) genotypes (**A1**). A higher delta mBI was observed in RRMS patients carrying the *ApaI* (CC) genotype vs. those with *ApaI* (AC) (*p* = 0.02) and (AA) (*p* = 0.005) genotypes (**A2**).

**Figure 3 ijms-24-13379-f003:**
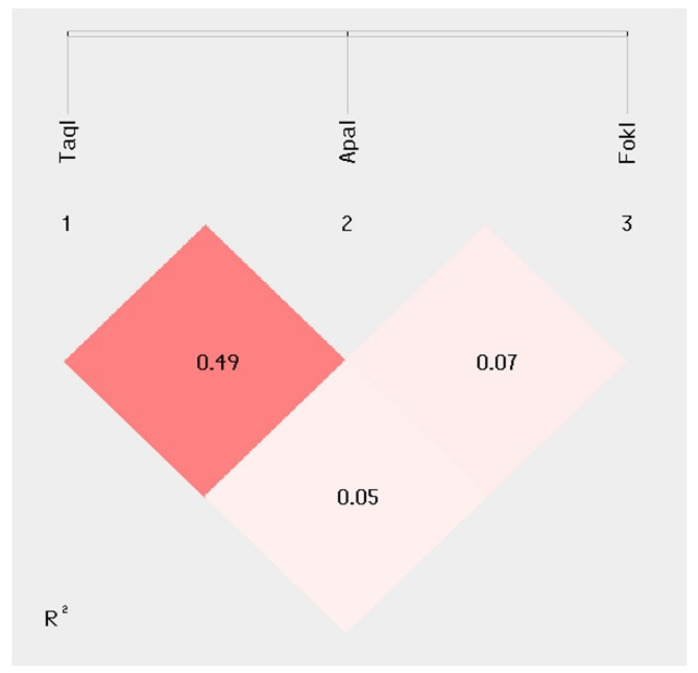
Linkage disequilibrium analysis (*r*^2^) for the SNPs (1) rs731236 (*TaqI*) (chr12: 46525024)*;* (2) rs7975232 (*ApaI*) (chr12: 46525104); (3) rs10735810 (*FokI*) (chr126559162).

**Table 1 ijms-24-13379-t001:** *VDR* genotype distribution in PPMS, SPMS, and RRMS patients.

*VDR*	PPMS		RRMS		SPMS		Total MS		*p*_c_ Value
rs731236 *TaqI*	N	%	N	%	N	%	N	%	
TT	15	41.7	29	30.9	41	34.5	85	34.1	
TC	18	50	38	40.4	56	47.1	112	45	
CC	3	8.3	27	28.7	22	18.5	52	20.9	
		HWE ns		HWE ns		HWE ns		HWE ns	ns
rs1544410 *BsmI*									
CC	15	41.7	31	33	40	33.6	86	34.5	
CT	16	44.4	32	34	45	37.8	93	37.3	
TT	5	13.9	31	33	34	28.6	70	28.1	
		HWE ns		HWE *p* < 0.01		HWE *p* < 0.01		HWE *p* < 0.001	ns
rs7975232 *ApaI*									
AA	7	19.4	34	36.2	40	33.6	80	32.1	
AC	20	55.6	42	44.7	57	47.9	119	47.8	
CC	9	25.0	18	19.1	22	18.5	50	20.1	
		HWE ns		HWE ns		HWE ns		HWE ns	ns
rs10735810 *FokI*									
CC	19	52.8	32	34.0	52	43.7	104	41.8	
CT	12	33.3	52	55.3	47	39.5	112	45	
TT	5	13.9	10	10.6	20	16.8	33	13.3	
		HWE ns		HWE ns		HWE ns		HWE ns	ns

HWE: Hardy–Weinberg equilibrium; *p*_c_: *p* value adjusted with Bonferroni method for multiple comparisons; ns: not statistically significant.

**Table 2 ijms-24-13379-t002:** EDSS, mBI, and pain NRS scores at the admittance (T0) and after MDR (T1) are reported as median and interquartile range (IQR). The Wilcoxon signed-rank test for repeated measures was performed in the overall group and in PMS and RRMS patients in order to compare T0 and T1 scores, and *p* values are reported in each row. EDSS: Expanded Disability Status Scale; mBI: Modified Barthel index; NRS: pain numerical rating score.

	EDSS T0		EDSS T1			mBI T0		mBI T1			Pain NRS T0		Pain NRS T1		
	Median	IQR	Median	IQR	*p* Value	Median	IQR	Median	IQR	*p* Value	Median	IQR	Median	IQR	*p* Value
**Total**	6.5	1.5	6.5	1.0	<0.001	65.0	27.0	75.0	26.0	<0.001	5.0	4.0	3.0	4.0	<0.001
**PMS (N = 155)**	7.0	1.5	6.5	1.3	<0.001	57.0	30.5	65.0	29.5	<0.001	5.0	4.0	3.0	4.0	<0.001
**RRMS (N = 94)**	6.0	1.5	6.0	1.5	<0.001	75.0	14.0	83.0	11.8	<0.001	5.0	3.0	3.0	4.0	<0.001

**Table 3 ijms-24-13379-t003:** General linear model of the correlation of delta mBi with *VDR TaqI* Dominant model, *ApaI* Recessive model, and *FokI* genotypes. Gender, mBI at baseline (T0), age, disease duration, days of hospitalization, number of interventions, and *DRB1*15.01* positivity are input as covariates.

Delta mBI in RRMS Patients	Beta Value	Standard Error	t Value	*p* Value
Intercept	35.91	6.75	5.32	<0.0001
Sex: Female vs. Male	0.03	1.45	0.02	0.9851
mBI T0 (1 unit more)	**−0.23**	**0.06**	**−3.94**	**0.0002**
Age (1 year more)	**−0.22**	**0.07**	**−3.06**	**0.003**
N of interventions (1 more)	−0.71	0.67	−1.06	0.2925
Disease duration (1 year more)	−0.01	0.07	−0.01	0.9897
hospitalization duration (1 day more)	−0.07	0.07	−0.88	0.3827
*DRB1*15.01* positivity	1.78	1.51	1.19	0.2379
*TaqI* TT vs. (TC + CC)	**6.35**	**1.65**	**3.86**	**0.0002**
*ApaI* CC vs. (AC + AA)	0.23	1.94	0.12	0.9064
*FokI* TC vs. CC	1.09	1.39	0.78	0.4357
*FokI* TT vs. CC	1.89	2.29	0.83	0.4096

Siginficant values are reproted in bold.

**Table 4 ijms-24-13379-t004:** *VDR* haplotype correlation with the delta mBI: quantitative trait haplotype analysis. Significant correlation are marked by: *.

		Haplotype Association with Delta mBI							
PMS N = 155	RRMS N = 94	Haplotype	*TaqI*	*ApaI*	*FokI*	PMS	*p* Value	RRMS	*p* Value
freq	freq					Beta Value		Beta Value	
0.21	0.31	*VDR-1*	T	C	C	0.23	ns	3.24 *	0.007 *
0.29	0.26	*VDR-2*	C	A	C	−1.37	ns	−2.18 *	0.04 *
0.23	0.08	*VDR-3*	T	C	T	−0.44	ns	2.18	ns
0.09	0.20	*VDR-4*	C	A	T	1.72	ns	−2.34	ns
0.13	0.04	*VDR-5*	T	A	C	−0.88	ns	−1.14	ns
0.03	0.08	*VDR-6*	T	A	T	1.49	ns	3.15	ns

**Table 5 ijms-24-13379-t005:** Demographic and clinical data of 36 primary progressive (PPMS), 119 secondary progressive (SPMS), and 94 relapsing remitting (RRMS) patients.

	PPMS		RRMS		SPMS		Total MS		*p* Value
N	36		94		119		249		
**Female:** **N (%)**	17 *	(47.2)	65 *	(69.1)	68	(57.1)	150	(60.2)	* = 0.02
**Age** **mean (SD)**	55.8 *	(12.2)	45.4 *°	(9.7)	53.7 °	(12.3)	50.8	(12.1)	* < 0.001 °<0.001
**DRB1*15 positive:** **N (%)**	7	(19.4)	23	(24.5)	36	(30.3)	66	(26.5)	ns
**disease duration** **Years’ median (IQR)**	16.0 ^	(12.5)	17.5 °	(13.0)	24.0 °^	(11.8)	20.0	(14.0)	^ < 0.001 ° < 0.001
**Hospitalization** **Days’ median (IQR)**	31	(14.3)	34	(13.0)	35	(12.5)	35.0	(13.0)	ns
**Interventions** **N (IQR)**	4	(1.5)	4	(2.0)	4	(2.0)	4	(2.0)	ns

N: absolute number; IQR: interquartile range; ns: not statistically significant. “*” comparison between PPMS and RRMS, “°” comparison between SPMS and RRMS, “^” comparison between PPMS and SPMS.

## Data Availability

The original contributions presented in the study can be find here: 10.5281/zenodo.8278839.

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
