# Peer review of "Vitamin D Receptor Gene Polymorphism Predicts the Outcome of Multidisciplinary Rehabilitation in Multiple Sclerosis Patients"

_ijms, 2023, doi:10.3390/ijms241713379_

Round 1

Reviewer 1 Report

Summary:

This study reports findings on three previously associated single nucleotide polymorphism previously associated with MS
in the VDR gene. They define different clinical MS phenotypic groups and then compare the scores for three different clinical measurement between subjects with different alleles at three SNPs of interests.  The find no differences in allele groups for two clinical measurements and do find differences for mBI between groups in one VDR allele. They confirm previous findings that the dominant allele for a VDR SNP termed Taql is correlated with more positive disease outcomes. The also make the novel finding that their treatment regiment has better results in subjects with the domain allele than the other alleles in the mBI measurement for the Taql allele.  

General comments:

The novelty of this study is the attempt to disease associated SNPs with clinical measures and clinical treatment outcomes. I believe this is important work. I believe the study design and the statistical testing are all sound as far as I can understand them.  However, the background, hypothesis, results and methods are all poorly presented and must be improved. Furthermore, in the study description the authors describe wanting to understand the “function” of the alleles.  I believe this is very misleading as the authors do not investigate the effect of the different alleles on the protein function of VDR, it might be previously described but it not within the current manuscript. The authors are just correlating the allele with the clinical outcome and this should be made clear.  The authors state in the abstract, this allele could be used as a biomarker, possibly to select subjects for treatment regiments.  I agree with this conclusion  and find it to be the most interesting finding in the manuscript, but this is not clearly discussed and does not come across in the paper.  It would be interesting for the authors to discuss more on the clinical uses of the biomarker and if there are treatment alternative that could be beneficial for those subjects with the non-dominant allele.

Specific concerns:

In the introduction:

-       The gene HLA-DRB1*15 should be more clearly defined in the intro.  The full name written out and the meaning or relevance of *15 (or *15.01 or *1501) nomenclature indicated. Later the authors mention DRB1*15 positive or negative.

-       What does it mean to be DRB1*15 positive or negative? Is this not a gene? Do they mean high or low expression or is this similar to a blood type marker?

-        When explaining the SNPs in the VDR gene the authors site a review.  Have these SNPs previously been associated with MS directly or only autoimmunity in general. In checking the review I find:

.. a study performed in 641 Italian MS patients and 558 HC showed a protective role against MS risk for the VDR TaqI T allele-HLA-DRB1*15+ haplotype 

This implies the DRB1*15 haplotype is the same as the Taql dominant allele.  The authors need to make the background genetics clear as their study depends on these gene and this allele.

-       The authors should explain why not meeting being in Hardy Weinberg Equilibrium mean a SNP should be excluded.

-       Line 119 the authors say the patients were “clustered”.  I think they mean grouped (by themselves) not by a clustering method as they don’t show any clustering.

-       Table 2 show the T0 and T1 scores for 3 different test and the result of the wilcoxin test between PMS and RRMS subject. What is the p-value in the total row?  Are separate wilcoxin test run to compare T0 PMS vs RRMS for each test (EDSS, mBI TO, Pain)? The authors indicated a repeated measure test is used, again indicating they are comparing the two time points matching patients.  The authors do not explain why a non-parametric test wa used, was the distribution tested for normality? They do refer to this test in the methods. This should be reformatted to make the comparisons clearer. In the current format it implies the p values are for T0 vs T1. Box plots  with significant differences indicated to accompany the table would also clarify the results.

-       Table 2 is cut off in the pdf file so I cannot see the Pain results.

-       Figure 1 the image is cut off in the and only the left and middle plots are visible. Figure 1 labels are hard to read. The figure legend doesn’t describe each panel within the figure. In the text the authors refer to “Figure 1 panel A3” and “Figure 1 B”.  The figure appears to be labelled as “1, 2, 3” in an upper row and “1, 2, 3” in a lower row with the titles “Relapsing Remitting MS patients –“ and “Progressive MS patients –“

-        The authors report differences in T0 between allele groups but only show the delta T0-T1 results in Figure 1. Data should be shown for all the comparisons described in the text.

-       Table 3 states : “Table 3: General linear model of correlation of delta mBi with VDR TaqI Dominant model, ApaI Recessive model and 169 FokI genotypes, adjusted for gender, mBI at baseline (T0) , age, disease duration, days of hospitalization, number 170 of intervention and DRB1*15 positivity as covariates.” Does this mean that Taql correlation was not adjusted for age or gender?

-       I am not clear what was being predicted by the logistic and or linear regressions.

-       In the methods only a Binary logistic regression analysis is referred to and the linear model and regression methods are not described.

-       No statement on data or code availability is provided. Raw data for test scores and haplotypes should be provided and analysis workbooks or at least a clearer description of the methods used should be shown. 

Author Response

Thank you very much for your useful critiques and suggestions, we answered to all you concerns  point by point

General comments:

The novelty of this study is the attempt to disease associated SNPs with clinical measures and clinical treatment outcomes. I believe this is important work. I believe the study design and the statistical testing are all sound as far as I can understand them.  However, the background, hypothesis, results and methods are all poorly presented and must be improved. Furthermore, in the study description the authors describe wanting to understand the “function” of the alleles.  I believe this is very misleading as the authors do not investigate the effect of the different alleles on the protein function of VDR, it might be previously described but it not within the current manuscript. The authors are just correlating the allele with the clinical outcome and this should be made clear.  

Thanks now we better clarified this point both in the abstract, introduction and in discussion section as follows:

Abstract:

 The aim of this study was to evaluate the association of the most important functional VDR SNPs (TaqI (T/C), ApaI (A/C), FokI (C/T) with functional outcome in MS patients undergoing a multidisciplinary inpatient rehabilitation (MDR), treatment, in order to  determine whether genetic profiling might be useful to identify subjects with a higher chance of recovery

Introduction

The ultimate aim was to ascertain whether such a genetic profiling could be useful in identifying subjects with a higher chance of recovery at the end of the MDR program.

Discussion:

In this study we investigated whether VDR gene polymorphisms could be considered as candidate biomarkers associated with  functional outcome after MDR rehabilitation in RRMS and PMS patients.

The authors state in the abstract, this allele could be used as a biomarker, possibly to select subjects for treatment regiments.  I agree with this conclusion  and find it to be the most interesting finding in the manuscript, but this is not clearly discussed and does not come across in the paper.  It would be interesting for the authors to discuss more on the clinical uses of the biomarker and if there are treatment alternative that could be beneficial for those subjects with the non-dominant allele.

We more in deep discussed this point in discussion section:

“ …an association between VDR genetic background and positive MDR outcome was found, although it is worth noting that any interpretation of our results for the purpose of individual patient profiling for rehabilitation programs is premature. Indeed, due to the characteristics of our inpatient MDR treatment, the activity undergone by each patient was heterogeneous as it was based on personalization and not on a pre-defined study protocol. Conversely, the latter was the case in other studies (23, 24) investigating the role of genetic biomarkers in MS rehabilitation. Giordano et al. (23) studied MS subjects undergoing inpatient motor rehabilitation and found an association of BDNF genetic variants with improvement of two clinical scales assessing ambulation and hand dexterity. On the other hand, Mori et al. (24) found an association between CB1 receptor polymorphisms and poor clinical benefit after physical therapy in an outpatient setting. Therefore, to the best of our knowledge, our findings mayprovide a novel piece of evidence on potential biomarkers predicting individualized MDR efficacy on functional autonomy in patients with MS.”

“ .. For these reasons, adding paraclinical information, such as genetic profiling, to better define intervention programs and priorities might represent a rewarding strategy.”

Specific concerns:

In the introduction:

-       The gene HLA-DRB1*15 should be more clearly defined in the intro.  The full name written out and the meaning or relevance of *15 (or *15.01 or *1501) nomenclature indicated. Later the authors mention DRB1*15 positive or negative.

We better defined the DRB1*15 with the right allelic nomenclature DRB1*15.01

-       What does it mean to be DRB1*15 positive or negative? Is this not a gene? Do they mean high or low expression or is this similar to a blood type marker?

Thanks we clarified in the text that DRB1*1501 positivity/negativity  means that the subjects do carry or not the allele HLA-DRB1*15.01  

“..subjects carrying the risk factor  DRB1*15 gene ((i.e. DRB1*1501 positive) “  

-        When explaining the SNPs in the VDR gene the authors site a review.  Have these SNPs previously been associated with MS directly or only autoimmunity in general. In checking the review I find: .. a study performed in 641 Italian MS patients and 558 HC showed a protective role against MS risk for the VDR TaqI T allele-HLA-DRB1*15+ haplotype 

This implies the DRB1*15 haplotype is the same as the Taql dominant allele.  The authors need to make the background genetics clear as their study depends on these gene and this allele.

Actually in that study the authors showed that in the group of subjects carrying DRB1*15 gene (that is the higher genetic risk factor for MS)  the co-presence of VDR TaqI T allele represents a protective factor, but this do not exclude that  subjects DRB1*15 positive may not  carry VDR TaqI T allele and vice versa that DRB1*15 negative subjects may carry VDR TaqI T allele because these two gene are not in linkage.

We better clarified this point in the text as follows:

An analysis of all the possible phenotypic combinations of the VDR genotype and the HLA-DRB1*15 allele in a cohort of MS patients and heathy controls showed that the co-presence of VDR TaqI TT in subjects carrying the risk factor  DRB1*15 gene ((i.e. DRB1*1501 positive)   was indeed significantly more frequent in healthy subjects compared to MS patients suggesting that VDR TaqI TT genotype may exert  a protective  role for MS (11)

-       The authors should explain why not meeting being in Hardy Weinberg Equilibrium mean a SNP should be excluded.

We explained this point as follow

Thus, since we cannot exclude that this disequilibrium could be due to due to a problematic genotype  assay (25), the BsmI polymorphism was excluded from all the subsequent analyses to avoid introducing possible correlation bias. 

-       Line 119 the authors say the patients were “clustered”.  I think they mean grouped (by themselves) not by a clustering method as they don’t show any clustering.

Thanks for your observation, we aplogize for the misuse of the terms and we have corrected the text as you suggested.

-       Table 2 show the T0 and T1 scores for 3 different test and the result of the wilcoxin test between PMS and RRMS subject. What is the p-value in the total row?

Are separate wilcoxin test run to compare T0 PMS vs RRMS for each test (EDSS, mBI TO, Pain)? The authors indicated a repeated measure test is used, again indicating they are comparing the two time points matching patients.  The authors do not explain why a non-parametric test wa used, was the distribution tested for normality? They do refer to this test in the methods. This should be reformatted to make the comparisons clearer. In the current format it implies the p values are for T0 vs T1. Box plots  with significant differences indicated to accompany the table would also clarify the results.

As you rightly underlined the results presented in Table 2 are quite confusing therefore we have separated the cross-sectional comparison of the scale scores at different time points in pMS and RRMS, from the longitudinal paired data analysis for repeated measures in each group of patients presented in Table 2.

Therefore, we added a figure as you suggested, which better illustrates the corss-sectional analysis and we better explained in the text the use of non parametric analysis and the results  as follows:

The EDSS, mBI and pain NRS scores resulted not normally distributed after  Kolmogorov Smirnov evaluation test, therefore their values were reported as median and interquartile range (IQR) and  the non-parametric Mann-Whitney comparison was adopted to evaluate the differences between PMS and RRMS patients at both  T0 and T1 points. Statistically higher EDSS scores were observed in PMS than in RRMS patients at both  T0 (median: 7 IQR:1,5 vs median 6 IQR:1,5 respectively) (p<0,001) and T1 (median: 6,5 IQR:1,3 vs  median: 6 IQR:1,5 respectively) (p<0,001) (Figure 1, panel A). On the other hand, PMS patients had lower mBI scores than RRMS  at both T0 (median:57 IQR:30,5 vs median:75 IQR:14,0)(p<0,0001) and at T1 (median:65,0 IQR:29,5 vs median:83,0 IQR:11,8)(p<0,0001) (Figure 1, panel B), confirming  an higher disability progression in PMS patients than in RRMS. Conversely, no difference was observed for pain NRS scores between groups (Figure 1, panel C).

In Table 2 we showed the repeated measures analysis 

The p-value in the total row refers to the  pairwise data comparison T0 vsT1 in the overall group of pMS+RRMS,   whereas the p-value reported in the singular rows refers to the pairwise data comparison in each group of patients (RRMS  and pMS).

We explained the results as follows:

The analysis of repeated measures of EDSS, mBI, pain NRS scores at T0 and T1 are shown in Table 2 , the results evidenced  a significant improvement of all outcome indicators after MDR in the whole groups of patients (Total row) p<0.001, and  in patients grouped according to disease phenotype (PMS and RRMS rows) p<0.001. No difference was observed between DRB1*15.01  positive and negative patients (data not shown). 

Than we rephrased the table legend as follows:

Table 2: EDSS, mBI and pain NRS scores at the admittance (T0) and   after MDR (T1) are reported as median and Interquartile range (IQR), analysis  for paired data was performed  in the overall group and in PMS and RRMS patients to compare T0 an T1 scores, and p value is reported in each row.

-       Table 2 is cut off in the pdf file so I cannot see the Pain results.

We apologyze for this , there was a problem with the final layout we took care of this inconvenience

-       Figure 1 the image is cut off in the and only the left and middle plots are visible. Figure 1 labels are hard to read. The figure legend doesn’t describe each panel within the figure. In the text the authors refer to “Figure 1 panel A3” and “Figure 1 B”.  The figure appears to be labelled as “1, 2, 3” in an upper row and “1, 2, 3” in a lower row with the titles “Relapsing Remitting MS patients –“ and “Progressive MS patients –“

We apologyze for this , there was a problem with the final layout we took care of this inconvenience 

We added panel references to the legend, adapted figure labels to make them easier to read and rephrased the figure legend to make  it self- explanatory

Figure 2: Delta mBI median distribution in relation to VDR TaqI, ApaI FokI genotypes in 94 RRMS (panel A)  and 155 PMS (panel B) patients. Higher delta mBI was observed in RRMS patients carrying the TaqI (TT) genotype than in those with the Taq (CT) (p<0,001) and Taq(CC) (p=0,001) genotypes (panel A1). Higher delta mBI was observed in RRMS patients carrying ApaI(CC) genotype vs those with ApaI (AC)(p=0,02) and (AA)(p=0,005) genotypes.

-        The authors report differences in T0 between allele groups but only show the delta T0-T1 results in Figure 1. Data should be shown for all the comparisons described in the text.

Thanks for your suggestion we have reported all the data evaluated at  T0 and T1 related to genotype distribution in PMS and RRMS patients in a supplementary table which is available but  does not overloadthe text with a large amount of data.

WE than rephrased the results as follows:

Genetic VDR pattern was evaluated in relationto EDSS, mBI and NRS values at admittance and with their changes after rehabilitation (Table 1s),), repeated measures analysis showed a significant variation of mBI in relation to  TaqI (p<0.001) and ApaI genotypes (p=0.01).

-       Table 3 states : “Table 3: General linear model of correlation of delta mBi with VDR TaqI Dominant model, ApaI Recessive model and 169 FokI genotypes, adjusted for gender, mBI at baseline (T0) , age, disease duration, days of hospitalization, number 170 of intervention and DRB1*15 positivity as covariates.” Does this mean that Taql correlation was not adjusted for age or gender?

All the polymorphisms correlation were imputed together in the same regression analysis and adjusted for all the covariates. We better described the model in the legend as follows:

General linear model of correlation of delta mBI with VDR  TaqI Dominant model,  ApaI Recessive model  and FokI  genotypes. Gender, mBI at baseline (T0), age, disease duration, days of hospitalization, number of interventions and  DRB1*15.01 positivity were imputed as covariates.

-       I am not clear what was being predicted by the logistic and or linear regressions.

Both regression analyses evaluated the association of mBI increase  with VDR polymorphisms using two different approaches: the first using delta mBI as the dependent continuous variable, the second by a more significant dicotomization of delta mBI increase.

We now specify in the text that:

Both the linear and the logistic regression analysis confirmed that mBI outcome after MDR was dependent on age, basal mBI and VDR TaqI genetic background.

In the methods only a Binary logistic regression analysis is referred to and the linear model and regression methods are not described.

We are  sorry that the  linear regression model was described in a confusing way. We have clarified the general linear regression analysis in the statistical section as follows:

A general linear regression model was then applied in RRMS and pMS separately,  considering as dependent variables delta mBI, delta EDSS and delta pain NRS which have been correlated  with VDR TaqI DM (TT vs TC+CC), ApaI (RM) (CC vs AC+AA), and FokI genotypes. In the same model, mBI, EDSS and pain NRS scores at the baseline (T0), age, years of disease duration, days of hospitalization and number of interventions were included as numerical  covariates, whereas VDR genotypes, DRB1*15.01 positivity and gender were imputed as categorical variables. 

-       No statement on data or code availability is provided. Raw data for test scores and haplotypes should be provided and analysis workbooks or at least a clearer description of the methods used should be shown. 

A description of the algorithm used to perform the haplotype analysis as reported by Shen et al (42) was added  to the statistical section as follows

To this end the SHESIS plus online software  http://analysis.bio-x.cn,  was adopted (40-41), that use an entropy-based algorithm to detect epistasis in the context of quantitative trait datasets (42)

Moreover, raw data have been loaded in the public repository Zenodo . The raw data supporting the main findings of this study are  available from the corresponding author upon reasonable request.

This information has been added at the end of the statistical section as:

Data Availability Statement

The original contributions presented in the study can be found here:   10.5281/zenodo.8278839

Reviewer 2 Report

Dear Authors,

The paper is well-written, and this is an interesting topic.

I have some minor comments and questions that I kindly put to the authors' consideration.

In the introduction and methods section, I would better explain what is "multidisciplinary rehabilitation." And if patients were exposed to a similar program or not. If not, this could possibly represent a limitation, which I would highlight in the discussion section.

What are the similarities and differences between your results and those previously published on the subject (REF 23 and 24)? Please add a few words in the discussion section.

Other than that, the paper is fine.

My decision is accepted with minor  revisions.

Author Response

Reviewer 2

Dear Authors,

The paper is well-written, and this is an interesting topic.

I have some minor comments and questions that I kindly put to the authors' consideration.

In the introduction and methods section, I would better explain what is "multidisciplinary rehabilitation." And if patients were exposed to a similar program or not. If not, this could possibly represent a limitation, which I would highlight in the discussion section.

Thanks for your suggestion, we have better specified in introduction section which is the multidisciplinary rehabilitation as follow:

Multidisciplinary rehabilitation (MDR) could be defined as an inpatient coordinated intervention, delivered by two or more disciplines based on a interdisciplinary evaluation assessing the patients’ needs (neurologist together with a physical medicine and rehabilitation specialist), which aims to helps a person to achieve and maintain maximal physical, psychological, social and vocational potential, and quality of life (QoL) consistent with impairment, environment and life goals. In principle, rehabilitation interventions for persons with MS may include exercise, functional training, equipment prescription, provision of assistive technology, orthotics prescription, teaching of compensatory strategies, caregiver/family support and education, counselling and referral to community resources. These can be targeted for a variety of impairments, such as mobility, fatigue, pain, dysphagia, bladder/bowel dysfunction, decreased independence in activities of daily living (ADL), communication, QoL, affective disorders and cognitive dysfunction. We have previously reported that inpatient MDR including multidomain interventions (motor, cognitive, speech, etc.) tailored on individual subject needs and goals

In the methods section we specified the percentage of subjects who underwent to each rehabilitation activity:

The intensive rehabilitation program included physiotherapy (i.e. motor rehabilitation) in all patients, associated with occupational therapy (in 78.3% of them), speech and swallowing rehabilitation (55,0%), cognitive rehabilitation (22.9%)., respiration therapy (16.5%), formal psychological counselling (28,5%) and physical therapy for pain (e.g. massage therapy, transcutaneous electrical nerve stimulation, electrical stimulation, ionophoresis) (89.1%)When needed, additional evaluations (cognitive, urological, ophthalmological, respiratory, etc.) were performed by the single discipline specialists to define the program. MDR consisted of daily individual sessions of one or more activities from Monday to Saturday for a total of at least 500 minutes a week, the duration of the admission was established following an intermediate multidisciplinary re-assessment of the program and goals with the involvement of physicians, therapists and nurses after two to three weeks of admission

What are the similarities and differences between your results and those previously published on the subject (REF 23 and 24)? Please add a few words in the discussion section.

Consistently with our previous report on a larger sample of subjects (22), including the subgroup genetically analysed for the present study, inpatient MDR was associated with improved autonomy in activities of daily living in a relevant proportion of  patients with MS. Moreover, an association was found between VDR genetic background and MDR positive outcome, although it is worth noting that any interpretation of our results for the purpose of individual patient profiling for rehabilitation programs is premature. Indeed, due to the characteristics of our inpatient MDR treatment, the activity undergone by single patients was heterogeneous because relying on a personalization and not on a pre-defined study protocol. Conversely, the latter was the case for other studies (23, 24) investigating the role of genetic biomarkers in MS rehabilitation. Giordano et al. (23) studied MS subjects undergoing inpatient motor rehabilitation and an association of BDNF genetic variants was found with the improvement of two clinical scales assessing ambulation and hand dexterity. On the other hand, Mori et al. (24) found a correlation between CB1 receptor polymorphisms and poor clinical benefit after physical therapy in an outpatient setting. To the best of our knowledge, our findings may, therefore, provide a novel piece of evidence about potential biomarkers predicting individualized MDR efficacy on functional autonomy in patients with MS.